# Influence of scar age, laser type and laser treatment intervals on adult burn scars: A systematic review and meta-analysis

Yangmyung Ma[1‡]*, Sabrina P. Barnes[2‡], Yung-Yi Chen[1], Naiem S. Moiemen[1,3,4], Janet M. Lord[1,4,5], Amanda V. Sardeli[1]

1 Institute of Inflammation and Ageing, University of Birmingham, Birmingham, United Kingdom, 2 Hull York Medical School, York, United Kingdom, 3 Department of Burns and Plastic Surgery, University Hospitals Birmingham NHS Foundation Trust, Birmingham, United Kingdom, 4 Scar Free Foundation Centre for Burns Research, University Hospitals Birmingham NHS Foundation Trust, Birmingham, United Kingdom, 5 National Institute for Health Research Surgical Reconstruction and Microbiology Research Centre, University Hospitals Birmingham NHS Foundation Trust, Birmingham, United Kingdom

‡ YM and SPB share co-first authors on this work.
* yangmyung.ma2@nhs.net

**Data Availability Statement:** All relevant data are within the manuscript and its Supporting Information files.

## Abstract

### Aim

The study aims to identify whether factors such as time to initiation of laser therapy following scar formation, type of laser used, laser treatment interval and presence of complications influence burn scar outcomes in adults, by meta-analysis of previous studies.

### Methods

A literature search was conducted in May 2022 in seven databases to select studies on the effects of laser therapy in adult hypertrophic burn scars. The study protocol was registered with PROSPERO (CRD42022347836).

### Results

Eleven studies were included in the meta-analysis, with a total of 491 patients. Laser therapy significantly improved overall VSS/POSAS, vascularity, pliability, pigmentation and scar height of burn scars. Vascularity improvement was greater when laser therapy was performed >12 months (-1.50 [95%CI = -2.58;-0.42], p = 0.01) compared to <12 months after injury (-0.39 [95%CI = -0.68; -0.10], p = 0.01), the same was true for scar height ((-1.36 [95%CI = -2.07; -0.66], p<0.001) vs (-0.56 [95%CI = -0.70; -0.42], p<0.001)). Pulse dye laser (-4.35 [95%CI = -6.83; -1.86], p<0.001) gave a greater reduction in VSS/POSAS scores compared to non-ablative (-1.52 [95%CI = -2.24; -0.83], p<0.001) and ablative lasers (-0.95 [95%CI = -1.31; -0.59], p<0.001).

### Conclusion

Efficacy of laser therapy is influenced by the time lapse after injury, the type of laser used and the interval between laser treatments. Significant heterogeneity was observed

**Funding:** The author(s) received no specific funding for this work.

**Competing interests:** The authors have declared that no competing interests exist.

among studies, suggesting the need to explore other factors that may affect scar outcomes.

## Introduction

Pathological scarring, such as hypertrophic scars, has a significant impact on a patient's quality of life. Complications following pathological scarring include contraction, reduction in range of movement, pruritus, pain, and discomfort [1]. In 2014, a literature review showed that 73% of patients with hypertrophic scarring experience pruritis and 68% experience pain [2]. These complications are often long-term, with research suggesting that the impact on the body's function, particularly after a major burn, can last beyond two years [3].

Treatment of pathological burns scars varies, either cosmetically, conservatively, or surgically. Laser therapy is a conservative method of treatment that offers a minimally invasive and low risk approach for the treatment of pathological burns scars. Laser type is classified into ablative carbon dioxide ($CO_2$) lasers, non-ablative fractional lasers and pulse dye lasers (PDLs). Ablative $CO_2$ lasers are used to reduce scar erythema for an improved visibility by targeting both dermal and epidermal layers of the skin, whereas non-ablative and fractional photothermolysis lasers address the thickness and volume of the scar by selectively damaging the dermis [4]. PDLs rely on a lower wave light frequency which is primarily absorbed by oxy-haemoglobin to improve scar vascularity and visibility [5]. All forms of lasers play an increasingly important role in burn scar management. However, there is variation in the efficacy of the treatment that may depend on the type of laser used, wavelength of laser and particularly on optimal timing for initiating laser therapy [4, 6].

The decision of how soon to begin laser therapy has depended upon scar maturation and other characteristics such as patient age, skin type, type of scar and co-morbidities. These factors are commonly used to predict treatment outcomes and prognosis [4]. However, other important factors such as optimal timing for initiation of laser therapy, laser types and treatment intervals for laser therapy have also been known to affect treatment outcomes, yet there is extensive heterogeneity within the literature surrounding the influence of these factors on outcomes after laser therapy [7]. Previously, optimal timing for laser therapy was once considered to be when the scar had reached full maturation. However, recent studies have suggested an association between early initiation and the decrease in symptoms, contractures, improvement in mobility and overall rehabilitation process, for example with the use of vascular devices in the months following burn or surgical injury [8, 9]. With evidence also suggesting that the incidence of adverse events of laser treatments is not affected by the age of scar at time of treatment [7], early laser treatment has become a potential method to minimise scar formation. Strengthening the evidence for factors that influence the efficacy of laser therapy would allow a more personalized and targeted treatment for the patient, depending upon scar maturation and patient characteristics, ultimately improving outcomes.

Recent meta-analyses have shown the efficacy of laser therapy on burn scars [10–13]. Although a positive outcome was observed in all studies, the individual studies only focused on one particular laser ($CO_2$) and observed significant heterogeneity in their data. No meta-analysis to date has considered the effects of optimal timing of laser therapy on burn scar outcomes in adults and thus this raises the possibility that this factor may be causing the heterogeneity.

In this way, the aim of this study was to identify the true effect of laser therapy on burn scar outcomes (VSS/POSAS scores, vascularity, pliability, pigmentation and scar height) through a comprehensive meta-analysis, considering the influence of different times to initiate

treatment, types of lasers, laser treatment interval, complications with laser therapy, and the controls used within studies. Through exploration of the effect of these factors, it will be possible to further optimise treatment protocols for laser therapy and provide personalised patient care.

This study focused on the adult population only, owing to differences in the physiological and pathological response to burn injuries in adults and children and potential different responses to laser therapy [14, 15].

## Methods

This review was reported according to the Preferred Reporting Items for Systematic Review and Meta-Analyses (PRISMA) guidelines.

### Protocol and registration

The study protocol was registered with PROSPERO (CRD42022347836).

### Eligibility criteria

The PICOS inclusion criteria were: (1) human adult patients (>18 years of age) with any post-burn hypertrophic scars; (2) undergoing interventions with laser therapy; (3) compared to themselves before treatment and/or a control group without laser therapy; (4) assessing objective scar measurement tools (e.g. via ultrasound guided measurement) and/or subjective Vancouver Scar Scale (VSS) / Patient and Observer Scar Assessment Scale (POSAS) scores, for pliability, pigmentation, vascularity, scar height (5), in retrospective, prospective or randomized control trial (RCT) studies. Only studies written in English or Chinese language were included. No date of publication restriction was applied.

### Exclusion criteria

The exclusion criteria for this study included: acne scars, surgical scars, articles published solely in abstract form (conference abstracts), article reviews, literature reviews, case reports and animal studies. Case reports were chosen for exclusion due to the underpowered nature of the study.

### Information sources

The databases accessed for the literature search included: PubMed, Google Scholar, EMBASE, Scopus, Cochrane Database of Systematic Reviews and University Library of York and Hull. All databases were accessed from 25th May 2022 to database inception.

### Search

The search strategy involved using pre-defined keywords with corresponding medical subject headings (MeSH) which included 'hypertrophic scar', 'cicatrix', 'keloid', 'scar', 'burn', 'major burn', 'thermal injury', 'severe burn', 'laser', 'laser therapy', 'ablative', 'pulse dye laser', 'ablation-therapy'. Scar, burn, and laser were all searched with truncation. Forwards and backwards citation searching as well as grey literature was checked to identify further articles.

### Study selection

All articles were downloaded onto Covidence, a programme used for primary screening and data extraction for researchers conducting standard intervention reviews. Duplicates were

deleted and the remaining articles were screened by two authors independently following pre-defined criteria. Full text of included studies were retrieved and further analysed independently, and any discrepancies concerning the articles' inclusion/exclusion was resolved through discussion from all authors. Articles written in Chinese were translated into English for inclusion in the title and abstract screening.

## Data collection process

Data extraction was completed by using a bespoke data extraction form. Data was extracted for the following categories: population (number of patients, age, scar age), intervention (laser type, number of treatments, treatment interval, scar assessment tools used), and outcomes of the study divided (overall VSS/POSAS scores, vascularity, pliability, pigmentation, scar height, complications). Two independent reviewers extracted the data from the studies and analysed the mean and standard deviation of before and after the 'early' and 'latent' period. Any discrepancies or disagreements with regards to data extraction were resolved through discussion with all authors.

For the purposes of the systematic review, the following terms were defined: 'laser ' as a scar therapy utilising photothermal energy to target intra and extra-cellular structures within the scar tissue [16], all types of lasers were included–ablative, PDL, non-ablative. 'Hypertrophic burn scars' were defined as pathological scarring due to major burns characterised by red, raised and rigid scar tissue that contracts and limits normal motion of the skin [17]. The age of scar was categorised into 'early' or 'latent', with 'early' being less than and including 12 months old and 'latent' being more than 12 months old.

## Risk of bias in individual studies

To determine the methodological quality and risk of bias of the included articles, full-text articles were assessed using the ROBINS-E tool for non-randomised studies of interventions and RoB tool for randomised controlled trials [18, 19]. These results were presented in Robvis format [20]. Two independent reviewers assessed the risk of bias and any discrepancies between the results were resolved by a third reviewer.

## Statistical analysis

The five meta-analyses, testing the effects of early and latent laser therapy using (1) overall scar improvement (assessed by VSS and POSAS in score points), (2) scar vascularity (score points), (3) scar pliability (score points), (4) scar pigmentation (score points), and (5) scar height (score points/nanometres) in burn scars of adult patients were performed using the Comprehensive Meta-Analysis (CMA) software version 3.3.070. The effect size was calculated based on the standard mean difference between before and after intervention (retrospective or prospective studies) or between differences in delta (before versus after) of control and intervention groups (RCTs). When there was no significant heterogeneity, fixed models were selected and when there was significant heterogeneity, random effects model was selected for analysis. Conservative pre-post correlations of 0.05 were assumed [21].

Subgroup analyses were conducted to explore confounding factors that could be influencing any heterogeneity in each of the five outcomes. The subgroup analyses considered the effects of characteristics of the study population, treatment methods and duration of the intervention on the main effects. The following subgroups were tested: Scar age (Early [<12 months] versus latent [>12 months] initiation of treatment), type of laser (ablative, PDL or non-ablative), interval length of laser treatment application (<4 weeks, 4–8 weeks, >8 weeks), presence or absence of complications reported (presence: bleeding, swelling,

hyperpigmentation, hypopigmentation, pain, blisters, pruritus, erythema, seepage and absence: no complications) and use of control group (with or without a control group). When an included study did not fit the category of subgroup or did not report the information, the study was excluded from that specific subgroup analysis. For all analyses, the p-value < 0.05 was considered significant. The Egger test was used to test the publication bias considering the p-value < 0.05.

## Results

A total of 2,955 papers were exported to Covidence software and were subject to inclusion and exclusion criteria to yield eleven papers that could be used for meta-analyses. Fig 1 presents this data in the flowchart of selection of the studies. Papers were excluded from the screening process if they were the wrong study design, comparator, patient population or intervention.

### Characteristics of the studies

The eleven studies included into the meta-analysis had a varied publication date from June 2009 to April 2022. The studies utilised a combination of study designs; five were RCTs and six were prospective studies [22–32]. A total of 491 participants were included in the 11 studies, and Tan et al. had the largest population size of 221 [29]. The studies were undertaken in five countries, with China being the most common location. The demographics reported showed an average patient age of 33.6 years with a 1:2 ratio of men to women. The studies used various lasers for the treatment method. Ablative $CO_2$ lasers were the most common, used in six studies at a frequency of 10,600nm. PDL was used in two studies, with the remaining three studies using non-ablative fractional lasers. The treatment duration, treatment interval and number of sessions varied between studies. The studies mostly relied on the VSS or POSAS as an outcome measure. Table 1 shows the characteristics of the included studies.

### Quality of studies

Six of the non-randomised studies scored an overall low risk of bias. Most prospective studies had some concerns with bias due to confounding. Five RCTs showed overall low risk of bias, and one with high risk. The RCT with the overall high risk was due to a high risk in one domain (bias arising from randomisation process). Figs 2 and 3 represents the risk of bias assessment for non-randomised studies and randomised studies respectively.

### Evidence synthesis

Our results showed that laser therapy significantly reduced VSS/POSAS scores (Fig 4A), vascularity (Fig 4B), pliability (Fig 4C), pigmentation (Fig 4D), and scar height (Fig 4E) in the overall analyses. Due to the presence of outliers in these meta-analyses, we tested the reliability of these results by analysis of one study removed, and the exact same mean and 95% CI were found for each of the five outcomes, reinforcing that no single study was impacting the overall results. There was no risk of publication bias for VSS/POSAS, pliability, pigmentation and scar height meta-analyses (2-tailed p-value of Egger test = 0.06, 0.13, 0.72, 0.11 respectively), however there was a significant risk of publication bias for the vascularity meta-analysis (2-tailed p-value of Egger test = 0.04). Table 2 shows the subgroup analyses for the outcomes tested.

Although both early (<12 months since injury) and latent (>12 months since injury) laser therapy were efficient at improving all outcomes investigated, latent laser therapy was more beneficial for vascularity and scar height than early treatment initiation. Ablative laser was the only laser type tested for vascularity, pliability and scar height outcomes and it significantly

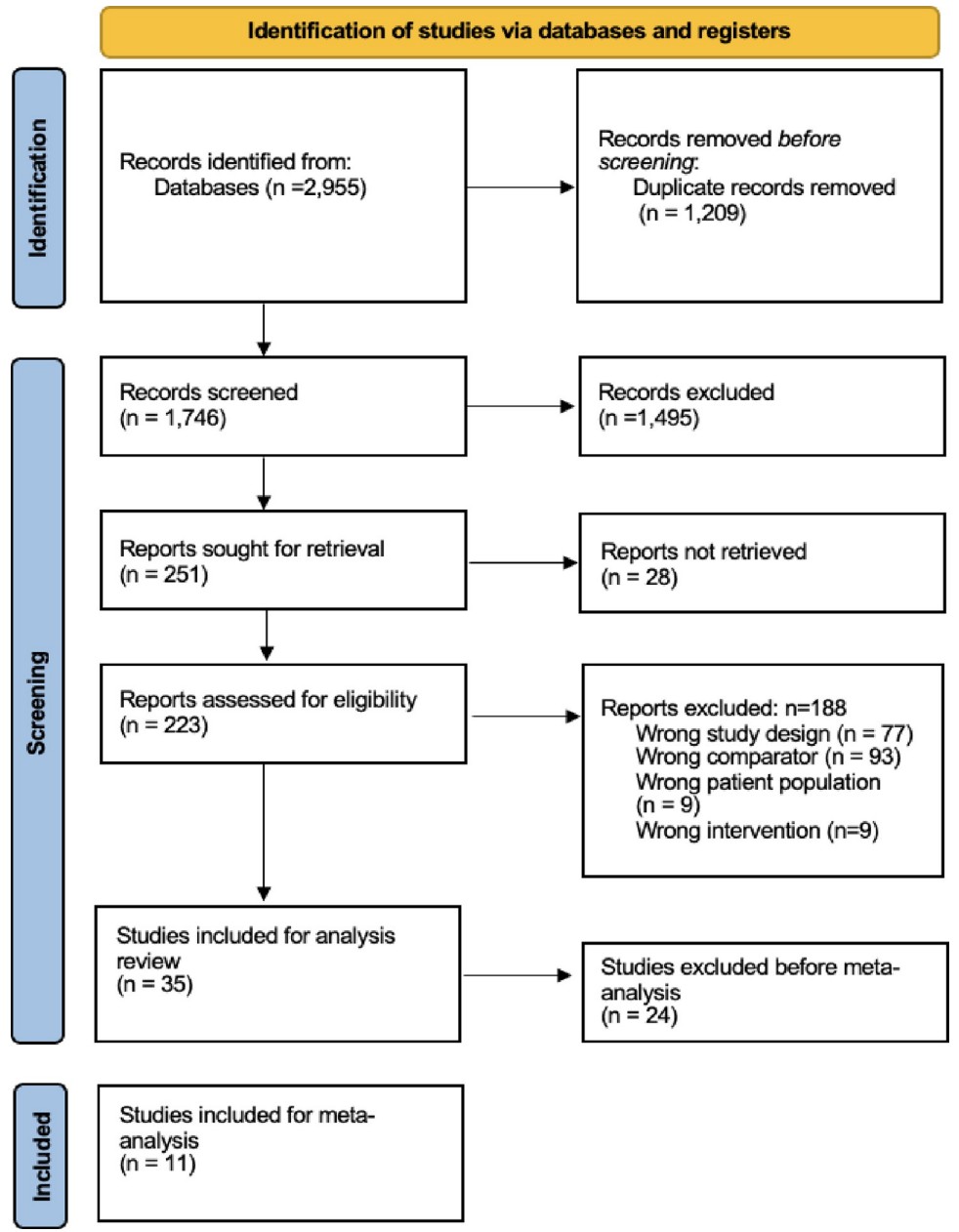

**Fig 1. Flowchart of selection of the studies.**

reduced these outcomes. Non-ablative lasers did not reduce pigmentation, whereas ablative lasers reduced this outcome significantly. For VSS/POSAS scores, significant differences were observed between the three types of lasers tested, where PDL was the most effective, compared to ablative and non-ablative lasers.

Shorter interval lengths between treatments were better than longer intervals for all the outcomes investigated, with the exception of pigmentation that had similar reduction for interval lengths of 4 to 8 weeks and >8 weeks. For VSS/POSAS scores, vascularity, pliability and scar height, a better response were seen for interval lengths of 4 to 8 weeks compared to >8 weeks and for VSS/POSAS scores, interval lengths of <4 weeks reduced scores more than intervals between 4 to 8 weeks.

**Table 1. Characteristics of the studies included.** $CO_2$ = Carbon dioxide, Er:YAG = Erbium-doped yttrium aluminium garnet, mths. = months, nm = nanometres, POSAS = Patient and Observer Scar Assessment Scale, PDL = Pulse Dye Laser, Pt = patients, VSS = Vancouver Scar Scale, wks. = weeks, yrs. = years, OSI = Overall Scar Improvement, Pigm = Pigmentation, Pli = Pliability, Vas = Vascularity, SH = Scar Height, STex = Scar Texture, Pru = Pruritus, Per = Perception, SEry = Scar Erythema, SElas = Scar elasticity, NR = Not reported, Hypo = Hypopigmentation, Hypr = Hyperpigmentation.

| Author, Publication Date (Ref) | Country (Study Designs) | No. of pts | Mean Age / Range (yrs.) | Scar Age Category: Range (mths.) | Laser Type (Wavelength) | Total no. of Sessions (Interval, wks.) | Time of Assessment (mths.) | Measurement Tools Used | Outcomes Reported | Complic-ations |
|---|---|---|---|---|---|---|---|---|---|---|
| Haedersdal, 2009 [22] | Denmark (RCT) | 17 | 37 | Latent: 60–120 | Non-Ablative Fractional Photothermolysis (1540nm) | 2–6, (interval 4) | At 1, 2 | POSAS, Volumetric Measure, Digital Pictures | OSI, Pigm, Vas, SH and STex | NR |
| Lin, 2011 [23] | United States (RCT) | 20 | 39 | Latent: 24–120 | Non-Ablative Fractional Photothermolysis (1540nm) | 4 (interval 2) | At 1, 3 | POSAS, Volumetric Measure, Digital Pictures | OSI, Pigm, Vas, SH and STex | SEry, hypo, pain, swelling, scabbing |
| Taudorf, 2015 [24] | Denmark (RCT) | 20 | 38 | Latent: 60–120 | Non-Ablative Er:YAG (1,540nm) | 2–6 (interval 4–6) | At 1, 3, 6 | PSOAS, Volumetric Measure, Biopsy | OSI, Pigm, Vas, SH and STex | SEry, swelling |
| Wang, 2015 [25] | China (Cohort) | 37 | 27.11 | Latent: 12–24 | PDL (500-600nm) | 2–6 (NR) | At 3 | VSS, Volumetric Measure, Digital Pictures | OSI, Vas, Pigm, Pli, SH and Scar Colour | SEry, blister, swelling, hypr |
| Weshahy, 2020 [26] | Egypt (RCT) | 15 | 38.95 ±8.55 | Latent:12–120 | $CO_2$ Laser (10,600nm) | 6 (NR) | At 2 | VSS, PSOAS, Volumetric Measure Digital Pictures, Biopsy | OSI, Vas, Pigm, Pli, SH, STex, Pain, Pru and Collagen Levels | NR |
| Lee, 2021 [27] | Korea (Cohort) | 40 | 36±17 | Latent: 24–60 | $CO_2$ Laser (10,600nm) | NR (interval 4–8) | Varied | VSS | OSI, Vas, Pigm, Pli and SH | SEry, blister, hypr, hypo |
| Li, 2021 [28] | China (Cohort) | 64 | 35.2 ±11.3 | Early: 6–12 | $CO_2$ Laser (10,600nm) | NR (interval 10) | At 8–12 | VSS | OSI, Vas, Pigm, Pli and SH | Swelling, pain, bleeding, pru, seepage |
| Tan, 2021 [29] | China (Cohort) | 221 | 33.6 | Early: 1–12 Latent: 12–24 | $CO_2$ Laser (10,600nm) | 1–4 (NR) | | VSS, Digital Pictures | OSI, Vas, Pigm, Pli, SH and Scar Colour | Swelling, bleeding, seepage |
| Xi, 2021 [30] | China (Cohort) | 16 | 27.5 | Latent: 12–24 | $CO_2$ Laser (10,600nm) | 6–12 (interval 8) | At 6 | VSS, Digital Pictures | OSI, Vas, Pigm, Pli, Height | NR |
| Yang, 2021 [31] | China (RCT) | 20 | 26 | Early: 1–3 | PDL (595nm) | NR (interval 1–4) | At 3 | VSS, Ultrasound | OSI, Vas, Pigm, Pli, SH and Thickness | NR |
| Ge, 2022 [32] | China (Cohort) | 21 | 31.4 | Early: 3–6 | $CO_2$ Laser (10,600nm) | Average number 4.86 ±1.74 (NR) | At 6–12 | POSAS, Ultrasound, Digital Pictures | OSI, Pigm, Vas, SH, STex, Pain, Pru and Per | NR |

Although laser therapy improved all outcomes in individuals with and without complications such as blistering, pain, bleeding, the studies isolating patients without complications tended to show higher reduction of overall VSS/POSAS scores and vascularity than studies including patients with complications.

Studies comparing the effects of laser within the same patient and comparing to an untreated area of scar as controls, tested only VSS/POSAS and pigmentation outcomes. Sensitivity analysis was conducted to investigate time-varying confounding which confirmed

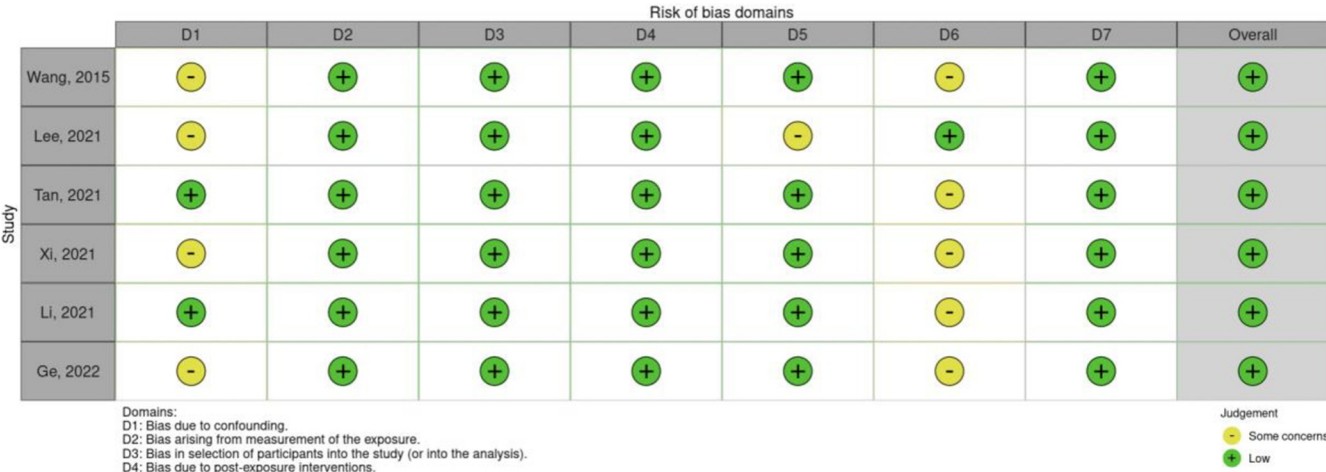

**Fig 2. Robvis–ROBINS-E assessment of bias for non-randomised studies.**

significant effects on the reduction of VSS/POSAS scores (-1.53 [-2.24; -0.83], p <0.001) with one study (Taudorf, 2015 [24]), whereas sensitivity analysis for these internally controlled studies (Haedersdal, 2009 [22]; Lin, 2011 [23]) on reduction of pigmentation did not lead to significant effects (-0.016 [-0.472; 0.440], p = 0.95) found in the overall analysis.

To infer about the clinical significance of these results, we ran sensitivity analysis of raw mean difference (RMD) for each scale, for each outcome. Laser therapy reduced near 3 points from the VSS scale (RMD -3.37 [-4.96; -1.78], p< 0.001, K = 8) as well as POSAS (RMD -3.19 [-4.14; -2.24], p < 0.001, K = 2). For the vascularity outcome, RMD for VSS points 0–5 was -2.35 ([-3.47; -1.24], p<0.001, K = 2), -1.55 for POSAS ([-2.15; -0.95], p<0.001, K = 1) and -0.45 for VSS points 0–3 ([-0.78; -0.12], p = 0.01, K = 5). RMD for pliability outcome showed a reduction of 1 for VSS points 0–5 (RMD -1.00 [-1.56; -0.44], p< 0.001, K = 6) and reduction of 1.68 for points 0–10 (RMD -1.68 [-2.27; -1.09], p< 0.001, K = 1). Laser therapy had a significant effect on pigmentation from VSS points 0–2 (RMD -0.276 [-0.366; -0.186], p< 0.001, K = 6) and points 0–10 (RMD -0.888 [-1.361; -0.415], p< 0.001, K = 2). For scar height, laser therapy reduced near 1 point from VSS scale 0–3 (RMD -0.96 [-1.33; -0.59], p< 0.001, K = 5) and reduction of 0.36mm via ultrasound (RMD -0.36 [-0.55; -0.17], p< 0.001, K = 2).

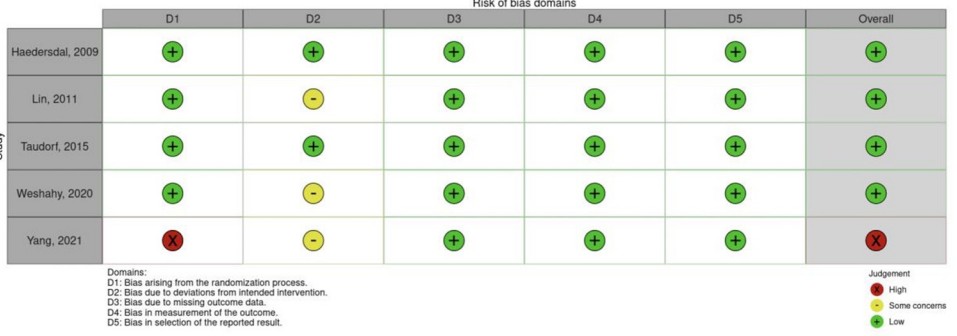

**Fig 3. Robvis–RoB assessment of bias for randomised studies.**

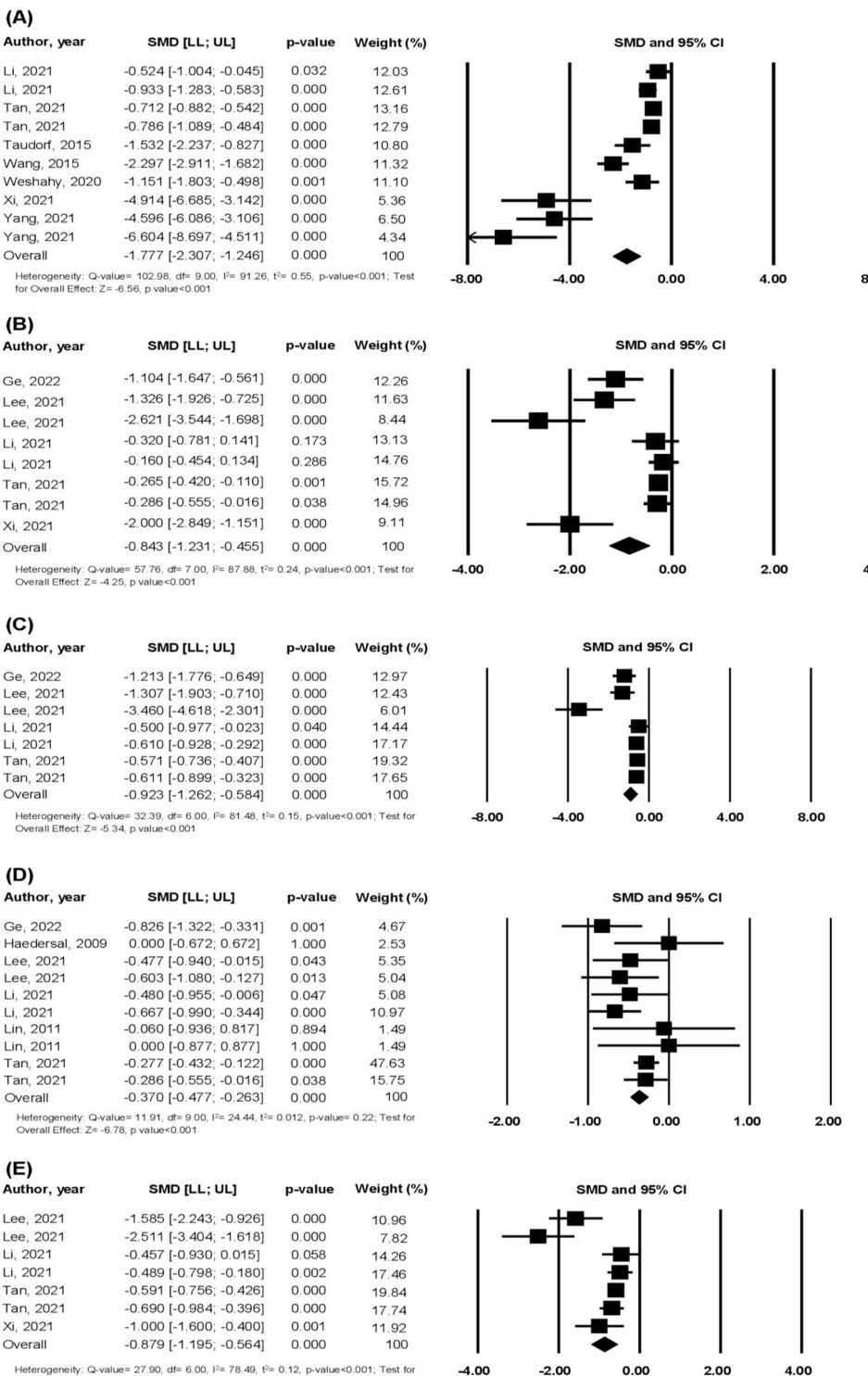

**Fig 4.** Forest Plots of the Effect of Laser Therapy on: (A) VSS/POSAS scores, (B) Vascularity, (C) Pliability, (D) Pigmentation, (E) Scar height. CI = confidence interval, LL = lower limit, POSAS = Patient and Observer Scar Assessment Scale, SMD = standardized mean difference, UL = upper limit, VSS = Vancouver Scar Scale.

**Table 2. Subgroup analysis of laser therapy on hypertrophic scars.**

| Subgroup | K | Study (reference) | SMD | LL and UL of 95% CI | p value within | p value between |
|---|---|---|---|---|---|---|
| **VSS / POSAS Total Scores** | | | | | | |
| Scar Age | | | | | | |
| Early | 5 | [28, 29, 31] | -1.85 | [-2.67 to -1.03] | <0.001 | 0.54 |
| Latent | 3 | [25, 29, 30] | -2.42 | [-4.07 to -0.77] | <0.001 | |
| Laser Type | | | | | | |
| Ablative | 6 | [26, 28–30] | -0.95 | [-1.31 to -0.59] | <0.001 | 0.01 |
| PDL | 3 | [25, 31] | -4.35 | [-6.83 to -1.86] | <0.001 | |
| Non-Ablative | 1 | [24] | -1.53 | [-2.24 to -0.83] | <0.001 | |
| Interval Length (weeks) | | | | | | |
| <4 | 2 | [31] | -5.46 | [-7.41 to -3.51] | <0.001 | <0.001 |
| 4 to 8 | 2 | [24, 30] | -3.12 | [-6.43 to 0.19] | 0.06 | |
| >8 | 2 | [28] | -0.76 | [-1.16 to -0.37] | <0.001 | |
| Complications | | | | | | |
| No | 4 | [26, 30, 31] | -4.22 | [-6.92 to -1.51] | <0.001 | 0.02 |
| Yes | 6 | [24, 25, 28, 29] | -1.05 | [-1.42 to -0.68] | <0.001 | |
| **Vascularity** | | | | | | |
| Scar Age | | | | | | |
| Early | 4 | [28, 29, 32] | -0.39 | [-0.68 to -0.10] | 0.01 | 0.05 |
| Latent | 3 | [27, 28, 30] | -1.50 | [-2.58 to -0.42] | 0.01 | |
| Interval Length (weeks) | | | | | | |
| 4 to 8 | 4 | [27, 30, 32] | -1.68 | [-2.30 to -1.05] | <0.001 | <0.001 |
| >8 | 2 | [28] | -0.21 | [-0.45 to 0.04] | 0.10 | |
| Complications | | | | | | |
| No | 2 | [30, 32] | -1.49 | [-2.36 to -0.62] | <0.001 | 0.08 |
| Yes | 4 | [27–29] | -0.63 | [-1.01 to -0.25] | <0.001 | |
| **Pliability** | | | | | | |
| Scar Age | | | | | | |
| Early | 4 | [28, 29, 32] | -0.64 | [-0.86 to -0.43] | <0.001 | 0.11 |
| Latent | 3 | [27, 28] | -1.66 | [-2.87 to -0.44] | 0.01 | |
| Interval Length (weeks) | | | | | | |
| 4 to 8 | 3 | [27, 32] | -1.86 | [-2.90 to -0.82] | <0.001 | 0.02 |
| >8 | 2 | [28] | -0.58 | [-0.84 to -0.31] | <0.001 | |
| Complications | | | | | | |
| No | 1 | [32] | -1.21 | [-1.78 to -0.65] | <0.001 | 0.33 |
| Yes | 6 | [27–29] | -0.88 | [-1.24 to -0.52] | <0.001 | |
| **Pigmentation** | | | | | | |
| Scar Age | | | | | | |
| Early | 4 | [28, 29, 32] | -0.39 | [-0.52 to -0.26] | <0.001 | 0.55 |
| Latent | 5 | [22, 23, 27, 29] | -0.32 | [-0.51 to -0.13] | 0.001 | |
| Laser Type | | | | | | |
| Ablative | 7 | [27–29, 32] | -0.39 | [-0.50 to -0.28] | <0.001 | 0.12 |
| Non-Ablative | 3 | [22, 23] | -0.02 | [-0.47 to 0.44] | 0.94 | |
| Interval Length (weeks) | | | | | | |
| <4 | 2 | [23] | -0.03 | [-0.65 to 0.59] | 0.93 | 0.24 |
| 4 to 8 | 4 | [22, 27, 32] | -0.54 | [-0.79 to -0.28] | <0.001 | |
| >8 | 2 | [28] | -0.61 | [-0.88 to -0.34] | <0.001 | |
| Complications | | | | | | |

*(Continued)*

**Table 2.** (Continued)

| Subgroup | K | Study (reference) | SMD | LL and UL of 95% CI | p value within | p value between |
|---|---|---|---|---|---|---|
| No | 2 | [22, 32] | -0.54 | [-0.93 to -0.14] | 0.008 | 0.40 |
| Yes | 8 | [23, 27–29] | -0.36 | [-0.47 to -0.25] | <0.001 | |
| **Scar Height** | | | | | | |
| Scar Age | | | | | | |
| Early | 3 | [28, 29] | -0.56 | [-0.70 to -0.42] | <0.001 | 0.03 |
| Latent | 4 | [27, 29, 30] | -1.36 | [-2.07 to -0.66] | <0.001 | |
| Interval Length (weeks) | | | | | | |
| 4 to 8 | 3 | [27, 30] | -1.64 | [-2.44 to -0.84] | <0.001 | 0.01 |
| >8 | 2 | [28] | -0.48 | [-0.74 to -0.22] | <0.001 | |
| Complications | | | | | | |
| No | 1 | [30] | -1.00 | [-1.60 to -0.40] | <0.001 | 0.71 |
| Yes | 6 | [27–29] | -0.87 | [-1.21 to -0.52] | <0.001 | |

CI = Confidence Interval, LL = Lower Limit, UL = Upper Limit, PDL = Pulse Dye Laser, POSAS = Patient and Observer Scar Assessment Scale, SMD = Standardized mean difference, VSS = Vancouver Scar Scale, K = Number of studies.

## Discussion

The exact mechanism of photothermolysis lasers on hypertrophic burn scars is currently unknown [13], but the theory relies on allowing new collagen to form in a controlled manner by causing either a photochemical reaction or heating to scars that have formed due to abnormal healing processes with increased collagen and fibronectin synthesis, fibroblast proliferation and neovascularisation [4]. Though the molecular and cellular mechanisms of scar formation for example through major involvement of matrix metalloproteinases and their inhibitors are well known, their effect and functions are not completely understood when they are induced by laser therapy. It is perhaps this lack of understanding that has led to several trials focussing on laser type, duration and optimal timing being conducted in an endeavour to minimise heterogeneity in outcomes [33, 34]. This meta-analysis aimed to address this heterogeneity by considering variables such as timing of treatment after injury, laser type, optimal spacing for laser intervention and complications.

Laser therapy offers a novel short term conservative treatment for burn scars [4]. Previous conservative methods, including silicone gel therapy and pressure garment therapy, lack extensive supporting evidence [35, 36]. For instance, silicone gel therapy is deemed 68% effective at reducing scar height whilst requiring high patient compliance and extensive treatment timelines [35]. Efficacy for pressure garment therapy requires application of this therapy for 23 hours per day for a miniumin of six months. This is an unrealistic expectation for patients especially in warmer climates, with well recognised complications of dermatitis [36]. Laser therapy on the other hand allows for minimal interaction for patients with health care in weekly sessions, whilst physiologically improving burn scars with minimal complications and evidence-based protocols [6].

In this analysis we included 11 studies, involving 491 patients that investigated five different outcomes of laser therapy on hypertrophic burn scars. This analysis was aimed to help clinicians and patients make evidence-based decisions particularly regarding optimal timing, type of laser and interval length of laser use when laser therapy is chosen as a method of scar management. The findings showed that laser remains an effective treatment for hypertrophic burn scars, and positive effects were observed when laser was used either before or after 12 months since injury.

Wound healing occurs in three discrete phases of inflammation, proliferation, and remodelling [37] and balance of the three phases may allow wounds to heal without excessive fibrosis. For example, the inflammatory phase comprises the release of cytokines and chemokines, as well as recruitment of fibroblasts and macrophages to restore the skin barrier. The inflammatory stage proceeds to the proliferation stage which can persist up to six weeks [38]. The remodelling phase occurs when the fibroblast differentiates into myofibroblasts that contract and decrease the wound size before entering the maturation phase that typically lasts until 12 months but has been known to mature beyond this time [37]. Perturbation of collagen production and collagenase synthesis leads to disorganised bundles of collagen cross-linked tightly creating a hypertrophic scar [39, 40]. It may then be intuitive to use lasers to target this process of disorganised growth in its early stages. For example, in 2018, a systematic review showed positive results for reducing cutaneous scar formation through laser intervention at three months post injury. The authors found significant improvement of the use of lasers in the inflammatory phase (lasers were applied immediately after or during wound closure), proliferation phase (laser applied mainly at time of suture removal) and improvement in the remodelling phase. However, some of the results of studies did not always reach significance and the population studied did not include patients with hypertrophic burn scars [41]. These results may well have influenced the adoption of early interventions with lasers in burns patients with hypertrophic scars, though our study does also support their use in more established scars.

Significant reduction of vascularity and scar height was observed with latent laser therapy, while no significant difference was found between early and latent laser therapy particularly in VSS/POSAS scores. This may be attributed to recent evidence which has shown that hypertrophic scars take significantly more time to completely mature than previously believed [42, 43]. A study in 2019 showed that mean maturation time for patients <30 years old was 35.76 months, 34.64 months for 30–55 year old patients and 22.53 months for >55-year-old patients. This suggests that the hypertrophic burn scars that were considered latent in this analysis may have been scars that have not fully matured and thus should have been considered and analysed in the early group.

Our subgroup analysis showed that laser type and the interval of laser use made a significant impact on the main results. The selection of laser depends on the principle that targeted tissue has a greater optical absorption at a specific wavelength compared to the surrounding tissue [4]. The subgroup analysis showed that PDL showed the greatest effect in improving VSS/POSAS scores. A recent retrospective study has shown the effectiveness of PDL, particularly in the early phases of wound healing, in optimising scar formation of hypertrophic burn scars [44]. However, the population of this study were children with Fitzpatrick skin type III and IV. PDLs work by targeting haemoglobin in blood vessels, resulting in selective photothermolysis, and they are generally considered safer than ablative lasers but have less penetration depth. PDL has been known to help reduce vascularity to reduce erythema, pruritis, pigmentation, hypertrophy and neuropathic pain from hypertrophic scars and can therefore be useful in the early stages of wound healing when the scar is thinner and more vascular [45–47].

In contrast, not much is known on the optimal interval for laser therapy with the need for long-term studies to be published to determine proper follow-up intervals [3]. Our results showed that shorter intervals helped significantly reduce VSS/POSAS scores, vascularity, pliability and scar height compared to intervals of >8 weeks. Recurrence is a main problem particularly with pathological keloid and hypertrophic burn scars with scar recurrence reported to present as early as two weeks and up to three years particularly following ablative laser therapy [48, 49]. Studies that used laser therapy at shorter intervals may have observed better outcomes owing to starting treatment before cellular and molecular processes for scar recurrence can occur.

Finally, we investigated whether any complications, such as blistering, bleeding etc, affected the main results. Studies that did not report any complications post laser therapy saw significantly reduced VSS/POSAS scores. Although a significant difference between studies with and without complications was observed in only one outcome, it would seem that the absence of complications post laser therapy may be indicative of improved scar outcomes.

The main limitation in this meta-analysis was the significant study heterogeneity. We have suggested the confounding factors that influence the main results, but other factors such as patient age, sex, skin type, co-morbidities and specific location of the burn scar on the body were not considered as they were not differentiated in the studies. Of particular note, the total number sessions was an important confounding factor that was not further analysed. This was due to the incomparability of results as most of the data provided was given as ranges by the individual studies. Another limitation is that laser interval and laser type subgroup analyses had limited data, with some of the results based on a single study. Analysis from a single study is not representative of the population and thus presents a selection bias. The small number of studies in these subgroup analyses also prevented further analysis of the data to isolate one outcome in a subgroup within another subgroup (e.g., comparing treatment interval outcomes within the types of laser treatments). It is important to note that subgroup analysis is a form of exploratory analysis with low level of evidence, as it is based on comparisons of various studies.

Significant results for sensitivity analysis of controls within studies was only available for VSS/POSAS scores in this study with only one study being tested. More controlled studies comparing laser therapy on the same patient and same scar is required to confirm whether scar improvement observed before and after laser therapy was an effect of laser therapy rather than an effect of time. In light of the small number of studies found for subgroup analyses, this affirms the need for further research to confirm the specific hypotheses raised within the subgroup analysis. Specifically, the authors advocate the need for future studies to investigate outcomes of laser therapy through comparison of different initiation times, type of laser therapies, and treatment intervals as well as investigating the long-term effects of laser therapy on scar recurrence. As such, the true effect of laser therapy may be further understood and used to guide safe clinical practice.

## Conclusion

Laser therapy is an effective method of management for hypertrophic burns scars, with either early or latent initiation. This perhaps suggests that initiation of laser therapy should be decided after consideration of the patients' factors and subsequently tailored. The type of laser and interval length between applications influences effectiveness whereby studies that used PDL observed the greatest improvement in VSS/POSAS scores and studies that used laser at shorter intervals observed the greatest improvement in VSS/POSAS scores, vascularity, pliability and scar height.

## Supporting information

**S1 Checklist. PRISMA 2020 checklist.**
(PDF)

**S2 Checklist. Meta-analysis on genetic association studies checklist.**
(DOCX)

## Author Contributions

**Conceptualization:** Naiem S. Moiemen, Janet M. Lord.

**Data curation:** Yangmyung Ma, Sabrina P. Barnes.

**Formal analysis:** Yangmyung Ma, Sabrina P. Barnes, Amanda V. Sardeli.

**Investigation:** Yangmyung Ma, Sabrina P. Barnes, Amanda V. Sardeli.

**Methodology:** Yangmyung Ma.

**Project administration:** Yangmyung Ma, Sabrina P. Barnes, Amanda V. Sardeli.

**Supervision:** Amanda V. Sardeli.

**Visualization:** Amanda V. Sardeli.

**Writing – original draft:** Yangmyung Ma, Sabrina P. Barnes, Amanda V. Sardeli.

**Writing – review & editing:** Yangmyung Ma, Sabrina P. Barnes, Yung-Yi Chen, Naiem S. Moiemen, Janet M. Lord, Amanda V. Sardeli.

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
