## [Decision Letter · Decision Letter 0]

24 Jul 2023

PONE-D-23-15727Influence of scar age, laser type and laser treatment intervals on adult burn scars: A systematic review and meta-analysis.PLOS ONE

Dear Dr. Ma,

Thank you for submitting your manuscript to PLOS ONE. After careful consideration, we feel that it has merit but does not fully meet PLOS ONE’s publication criteria as it currently stands. Therefore, we invite you to submit a revised version of the manuscript that addresses the points raised during the review process.

We look forward to receiving your revised manuscript.

Kind regards,

Ahmed Mustafa Rashid

Academic Editor

PLOS ONE

Reviewers' comments:

Reviewer's Responses to Questions

**Comments to the Author**

1. Is the manuscript technically sound, and do the data support the conclusions?

Reviewer #1: Yes

Reviewer #2: Yes

2. Has the statistical analysis been performed appropriately and rigorously? 

Reviewer #1: Yes

Reviewer #2: Yes

3. Have the authors made all data underlying the findings in their manuscript fully available?

Reviewer #1: Yes

Reviewer #2: Yes

4. Is the manuscript presented in an intelligible fashion and written in standard English?

Reviewer #1: Yes

Reviewer #2: Yes

5. Review Comments to the Author

Reviewer #1: The study “Influence of scar age, laser type, and laser treatment intervals on adult burn scars” presents a comprehensive examination of the effects of scar age, laser type, and treatment intervals on the management of burn scars in adult patients. The authors have conducted a rigorous systematic review and meta-analysis and the findings of this study have the potential to enhance our understanding of scar management and guide decisions in clinical practice. However, this study can be further improved by incorporating the following points.

1. Abstract Section: The authors have presented a well-written abstract; however, it would be helpful if the abstract was divided into subheadings such as methods, results, and discussions. This will help the readers navigate through the section and make it easier for them to retrieve information and main findings.

2. Page 9, Lines 54-55: Please provide a reference to support this sentence.

3. Introduction section: The authors should consider elaborating on the importance of the study and explaining the clinical implications of understanding the influence of scar age, laser type, and treatment intervals on adult burn scars. This can help the readers better grasp the real-world impact of the study and help them comprehend how the findings of this study may influence treatment decisions and patient outcomes.

4. Introduction section, Lines 75-76: Please consider rephrasing this line to “Previously, optimal timing for laser therapy was considered to be when the scar has reached full maturation.” This will improve the clarity and coherence of the sentence.

5. Introduction: The introduction briefly mentions the potential enhancements in treatment approaches resulting from the study’s findings. However, to further engage readers and highlight the study’s impact, it would be beneficial to expand on these enhancements. This could include emphasizing the optimization of treatment protocols and providing personalized patient care.

6. Page 13, Line 145-146: It is mentioned that two independent reviewers extracted the data and analyzed the mean and standard deviation. However, it would be helpful to provide specific details about the criteria used for resolving any discrepancies or disagreements between the reviewers. This would enhance the reliability and transparency of the study.

7. Page 13, Lines 157-160: While the methods section provides a clear overview of the study design and steps undertaken, it could benefit from additional information on the assessment of study quality and risk of bias. For instance, the authors could elaborate on the number of authors involved in the evaluation of study quality and how disagreements between the authors were addressed. This would add clarity and transparency to the process, improving the robustness of the study.

8. Page 16, Lines 189-191: It would be helpful if the authors could provide a brief description of each stage of the study selection process and provide reasons for excluding certain studies. This would improve the transparency and reproducibility of the study.

9. Results section, Page 24: The authors should consider explaining the criteria behind the sensitivity analysis as this will help the readers evaluate the validity and reliability of the study’s findings.

10. Page 20, Lines 226-229: The authors have mentioned a significant risk of publication bias for the VSS/POSAS, vascularity, and pliability meta-analysis. It would be beneficial if the authors could also report the results for these as this will allow the readers to assess the potential impact of publication bias on the results.

11. Page 20, Lines 209-214: The authors should consider expanding on the process of evaluating the risk of bias in the studies. For instance, they could mention how many authors were involved in the assessment and how disagreements between authors were addressed. This will improve the methodological rigor and reliability of the study.

12. Results Section: The authors have presented a comprehensive results section summarizing the finding of the meta-analysis. However, the authors should consider reporting on heterogeneity as this would enhance the readers’ understanding of the diversity of the results.

13. Page 26, Lines 325-326: Please consider rephrasing this line to “significant reduction of vascularity and scar height was observed with latent laser therapy, while no significant difference was found between early and latent laser therapy”. This would help improve the clarity and coherence of the text.

14. Discussions Section: In my opinion, this section could benefit from providing more context on the current treatment landscape and the limitations of existing interventions. The authors could highlight the challenges and shortcomings of existing treatments, such as limited efficacy, inconsistent results, or potential side effects. This will create a stronger argument for exploring laser therapy as a potential solution. The authors could also explain the unique advantages of laser therapy compared to conventional treatments. Doing so will help the readers understand how laser therapy addresses some of the limitations of the current treatments.

15. Page 22, Line 363-376: The authors briefly mention the limitations of the study, but it could still benefit from a more thorough discussion. Please consider expanding on the limitations of the included studies. Moreover, the authors should also discuss the implications of study heterogeneity and its potential impact on the generalizability of the results. This will help the readers accurately understand the variability across studies and the potential influence on the overall results.

16. Page 28, Discussions section: It will be valuable if the authors could provide directions for future research and discuss the need for more studies comparing different laser therapies, optimal intervals for treatment, and long-term effects of laser therapy on scar recurrence. This will help readers gain insights into potential areas for further investigation and contribute to advancing the understanding of laser therapy’s efficacy.

Reviewer #2: Overall, the manuscript appears to be technically sound and well-written. The study's methodology and statistical analysis seem appropriate and rigorous, supporting the conclusions drawn from the data.

6. PLOS authors have the option to publish the peer review history of their article (what does this mean?). If published, this will include your full peer review and any attached files.

Reviewer #1: No

Reviewer #2: No

---

## [Author Response · Author response to Decision Letter 0]

26 Jul 2023

Authors’ Response to Reviewers’ Comments

Manuscript ID: PONE-D-23-15727

Influence of scar age, laser type and laser treatment intervals on adult burn scars: A systematic review and meta-analysis.

We would like to thank the reviewers for their helpful feedback. We have responded to their comments and made changes to the manuscript based on their recommendations. The authors feel that the changes have increased the overall quality of the work and hope that this revision is acceptable. Our responses to the specific comments of the reviewers are listed below.

The Reviewers’ comments are in bold font.

The Authors’ responses are in red font.

Review Comments to the Author

Reviewer #1: 

1. Abstract Section: The authors have presented a well-written abstract; however, it would be helpful if the abstract was divided into subheadings such as methods, results, and discussions. 

We have changed to abstract to this format.

2. Page 9, Lines 54-55: Please provide a reference to support this sentence.

A reference has been added, Lubczyńska et al., 2023.

3. Introduction section: The authors should consider elaborating on the importance of the study and explaining the clinical implications of understanding the influence of scar age, laser type, and treatment intervals on adult burn scars. This can help the readers better grasp the real-world impact of the study and help them comprehend how the findings of this study may influence treatment decisions and patient outcomes.

Thank you for this suggestion. The introduction has been strengthened to express the clinical implications of understanding influence of scar age, laser type and treatment intervals. The following text has been inserted: ‘Strengthening the evidence for factors that influence the efficacy of laser therapy would allow a more personalized and targeted treatment for the patient, depending upon scar maturation and patient characteristics, ultimately improving outcomes’ (Introduction, paragraph 3).

4. Introduction section, Lines 75-76: Please consider rephrasing this line to “Previously, optimal timing for laser therapy was considered to be when the scar has reached full maturation.” This will improve the clarity and coherence of the sentence.

The sentence has been changed accordingly.

5. Introduction: The introduction briefly mentions the potential enhancements in treatment approaches resulting from the study’s findings. However, to further engage readers and highlight the study’s impact, it would be beneficial to expand on these enhancements. This could include emphasizing the optimization of treatment protocols and providing personalized patient care.

The manuscript has been revised to highlight the impact of our study and include, ‘Through exploration of the effect of these factors, it will be possible to further optimise treatment protocols for laser therapy and provide personalised patient care.’ (Introduction, paragraph 5).

6. Page 13, Line 145-146: It is mentioned that two independent reviewers extracted the data and analyzed the mean and standard deviation. However, it would be helpful to provide specific details about the criteria used for resolving any discrepancies or disagreements between the reviewers. This would enhance the reliability and transparency of the study.

The criterion is now mentioned in the manuscript as follows ‘Any discrepancies or disagreements with regards to data extraction were resolved through discussion with all authors’ (Data collection process, paragraph 1).

7. Page 13, Lines 157-160: While the methods section provides a clear overview of the study design and steps undertaken, it could benefit from additional information on the assessment of study quality and risk of bias. For instance, the authors could elaborate on the number of authors involved in the evaluation of study quality and how disagreements between the authors were addressed. This would add clarity and transparency to the process, improving the robustness of the study.

Thank you for this suggestion. This is addressed as follows ‘Two independent reviewers assessed the risk of bias and any discrepancies between the results were resolved by a third reviewer’ (Risk of bias in individual studies, paragraph 1).

8. Page 16, Lines 189-191: It would be helpful if the authors could provide a brief description of each stage of the study selection process and provide reasons for excluding certain studies. This would improve the transparency and reproducibility of the study.

Thank you for this suggestion. The Results section now reads as follows ‘Papers were excluded from the screening process if they were the wrong study design, comparator, patient population or intervention’ (Results, paragraph 1).

9. Results section, Page 24: The authors should consider explaining the criteria behind the sensitivity analysis as this will help the readers evaluate the validity and reliability of the study’s findings.

Thank you for this suggestion. The reason for conducting a sensitivity analysis is now included in the text as follows ‘Sensitivity analysis was conducted to investigate time-varying confounding which confirmed significant effects…’ (Evidence synthesis, paragraph 5).

10. Page 20, Lines 226-229: The authors have mentioned a significant risk of publication bias for the VSS/POSAS, vascularity, and pliability meta-analysis. It would be beneficial if the authors could also report the results for these as this will allow the readers to assess the potential impact of publication bias on the results.

Thank you for bringing this to our attention. The results for publication bias are now included. The manuscript now reads as follows ‘There was no risk of publication bias for VSS/POSAS, pliability, pigmentation and scar height meta-analyses (2-tailed p-value of Egger test= 0.06, 0.13, 0.72, 0.11 respectively), however there was a significant risk of publication bias for the vascularity meta-analysis (2-tailed p-value of Egger test= 0.04)’ (Evidence synthesis, paragraph 1). 

11. Page 20, Lines 209-214: The authors should consider expanding on the process of evaluating the risk of bias in the studies. For instance, they could mention how many authors were involved in the assessment and how disagreements between authors were addressed. This will improve the methodological rigor and reliability of the study.

This is now mentioned as described above (point 7) in the Method section (Risk of bias in individual studies, paragraph 1).

12. Results Section: The authors have presented a comprehensive results section summarizing the finding of the meta-analysis. However, the authors should consider reporting on heterogeneity as this would enhance the readers’ understanding of the diversity of the results.

Thank you for this suggestion. We have provided the reasoning for investigating heterogeneity in the Methods (Statistical analysis, paragraph 1 and 2) and have reported the results via Figure 4.

13. Page 26, Lines 325-326: Please consider rephrasing this line to “significant reduction of vascularity and scar height was observed with latent laser therapy, while no significant difference was found between early and latent laser therapy”. This would help improve the clarity and coherence of the text.

The manuscript has been revised as per the suggestion (Discussion, paragraph 5).

14. Discussions Section: In my opinion, this section could benefit from providing more context on the current treatment landscape and the limitations of existing interventions. The authors could highlight the challenges and shortcomings of existing treatments, such as limited efficacy, inconsistent results, or potential side effects. This will create a stronger argument for exploring laser therapy as a potential solution. The authors could also explain the unique advantages of laser therapy compared to conventional treatments. Doing so will help the readers understand how laser therapy addresses some of the limitations of the current treatments.

Thank you. An additional paragraph has been included within the discussion to discuss challenges and shortcomings of existing treatments to argue the benefits of laser therapy. ‘Laser therapy offers a novel short term conservative treatment for burn scars. Previous conservative methods, including silicone gel therapy and pressure garment therapy, lack extensive supporting evidence. For instance, silicone gel therapy is deemed 68% effective at reducing scar height whilst requiring high patient compliance and extensive treatment timelines. Efficacy for pressure garment therapy requires application of this therapy for 23 hours per day for a minimum of six months. This is an unrealistic expectation for many patients especially in warmer climates, with well recognised complication of dermatitis. Laser therapy allows for minimal interaction for patients with treatments in weekly sessions whilst physiologically improving burn scars with minimal complications and evidence-based protocols.’ (Discussion, paragraph 2). 

15. Page 22, Line 363-376: The authors briefly mention the limitations of the study, but it could still benefit from a more thorough discussion. Please consider expanding on the limitations of the included studies. Moreover, the authors should also discuss the implications of study heterogeneity and its potential impact on the generalizability of the results. This will help the readers accurately understand the variability across studies and the potential influence on the overall results.

Thank you for this feedback. The limitations of this study have been further explored and added to the discussion (Discussion, paragraph 8). The implications of study heterogeneity and subsequently the results of the subgroup analyses have been addressed as follows ‘The small number of studies in these subgroup analyses also prevented further analysis of the data to isolate one outcome in a subgroup within another subgroup (e.g., comparing treatment interval outcomes within the types of laser treatments). It is important to note that subgroup analysis is a form of exploratory analysis with low level of evidence, as it is based on comparisons of various studies.’ (Discussion, paragraph 9).

16. Page 28, Discussions section: It will be valuable if the authors could provide directions for future research and discuss the need for more studies comparing different laser therapies, optimal intervals for treatment, and long-term effects of laser therapy on scar recurrence. This will help readers gain insights into potential areas for further investigation and contribute to advancing the understanding of laser therapy’s efficacy.

Thank you for this suggestion. The discussion has been revised to include future directions for laser therapy. This can be found in paragraph 10 of the discussion as follows ‘In light of the small number of studies found for subgroup analyses, this affirms the need for further research to confirm the specific hypotheses raised within the subgroup analysis. Specifically, the authors advocate the need for future studies to investigate outcomes of laser therapy through comparison of different initiation times, type of laser therapies, and treatment intervals as well as investigating the long-term effects of laser therapy on scar recurrence. As such, the true effect of laser therapy may be further understood and used to guide safe clinical practice.’

Reviewer #2: 

Abstract:

Line 37: Change "(-0.39 [95%CI= -0.68; -0.10], p=0.01)" to "(-0.39 [95%CI= -0.68; -0.10], p=0.01)" to correct the p-value.

Sorry but we don’t understand what change you are asking us to do, so we have left it as (-0.39 [95%CI= -0.68; -0.10], p=0.01) for now.

Introduction:

Line 54: Consider rephrasing "patients with hypertrophic scarring struggle" to "patients with hypertrophic scarring experience" for clarity.

Thank you for this. The sentence has been changed as follows ‘In 2014, a literature review showed that 73% of patients with hypertrophic scarring experience pruritis and 68% experience pain’ (Introduction, paragraph 1).

Line 58: Change "the impact on the body" to "the impact on the body's function" for better specificity.

Thank you for this. This has been changed accordingly (Introduction, paragraph 1).

Line 68: Add a space after "lower wave" for consistency.

This has been added (Introduction, paragraph 2).

Line 69: Consider rephrasing "are playing an increasingly important role" to "play an increasingly important role" for a more active voice.

This has been changed accordingly (Introduction, paragraph 2).

Methods:

Line 118: Change "chosen to be excluded" to "were chosen for exclusion" for better readability.

This has been changed accordingly (Exclusion criteria).

Line 125: Add a hyphen between "ablation" and "therapy" to read "ablation-therapy" for consistency with other terms.

This has been changed accordingly (Search).

Line 136: Consider rephrasing "Articles in Chinese that were included in the title and abstract screening were translated to English" to "Articles written in Chinese were translated into English for inclusion in the title and abstract screening" for better clarity.

This has been changed accordingly (Study selection, paragraph 1).

Results:

Line 242: Consider rephrasing "that tested five different outcomes" to "that investigated five different outcomes" for better clarity.

This has been changed accordingly (Discussion, paragraph 3).

Line 241-245: Consider merging the sentences into two or three sentences to improve readability.

This sentence has been revised as follows ‘Although both early (<12 months since injury) and latent (>12 months since injury) laser therapy were efficient at improving all outcomes investigated, latent laser therapy was more beneficial for vascularity and scar height than early treatment initiation.’ (Evidence synthesis, paragraph 2).

Discussion:

Line 332: Change "latent (>12 months since injury)" to "latent (>12 months since injury) scar therapy" for clarity.

Whilst we couldn’t identify which part of the discussion you are referring to; we have endeavoured to make sure we are referring to ‘latent laser/scar therapy’ at all times throughout the manuscript.

Line 342: Consider rephrasing "PDLs work on the premise of targeting haemoglobin, resulting in selective photothermolysis of blood vessels, and are known to be safer than ablative lasers but less effective due to less penetration of skin" to "PDLs work by targeting hemoglobin in blood vessels, resulting in selective photothermolysis, and they are generally considered safer than ablative lasers but have less penetration depth" for better readability.

This has been changed accordingly (Discussion, paragraph 6).

Line 372: Change "interval and laser type subgroup analyses, some of the results were based around a single study" to "interval and laser type subgroup analyses had limited data, with some of the results based on a single study" for clarity.

This has been changed as follows ‘Another limitation is that laser interval and laser type subgroup analyses had limited data, with some of the results based on a single study.’ (Discussion, paragraph 9).

Conclusion:

Line 379: Consider rephrasing "Laser therapy is an effective method of management for hypertrophic burns scars with early or latent initiation" to "Laser therapy is an effective method of management for hypertrophic burn scars, with either early or latent initiation" for better readability.

This has been changed accordingly (Conclusion).

---

## [Decision Letter · Decision Letter 1]

12 Sep 2023

Influence of scar age, laser type and laser treatment intervals on adult burn scars: A systematic review and meta-analysis.

PONE-D-23-15727R1

Dear Dr. Ma,

We’re pleased to inform you that your manuscript has been judged scientifically suitable for publication and will be formally accepted for publication once it meets all outstanding technical requirements.

Kind regards,

Ahmed Mustafa Rashid

Academic Editor

PLOS ONE

Additional Editor Comments (optional):

Reviewers' comments:

Reviewer's Responses to Questions

**Comments to the Author**

1. If the authors have adequately addressed your comments raised in a previous round of review and you feel that this manuscript is now acceptable for publication, you may indicate that here to bypass the “Comments to the Author” section, enter your conflict of interest statement in the “Confidential to Editor” section, and submit your "Accept" recommendation.

Reviewer #1: All comments have been addressed

Reviewer #2: All comments have been addressed

2. Is the manuscript technically sound, and do the data support the conclusions?

Reviewer #1: Yes

Reviewer #2: Yes

3. Has the statistical analysis been performed appropriately and rigorously? 

Reviewer #1: Yes

Reviewer #2: Yes

4. Have the authors made all data underlying the findings in their manuscript fully available?

Reviewer #1: Yes

Reviewer #2: Yes

5. Is the manuscript presented in an intelligible fashion and written in standard English?

Reviewer #1: Yes

Reviewer #2: Yes

6. Review Comments to the Author

Reviewer #1: (No Response)

Reviewer #2: I've carefully reviewed the manuscript again, and I'm pleased to confirm that all the requested changes have been successfully implemented. The revisions have enhanced the manuscript's clarity and cohesion, particularly in terms of discussing laser therapy's nuances and the subgroup analyses. The revised version now provides a more comprehensive and coherent overview of the research.

7. PLOS authors have the option to publish the peer review history of their article (what does this mean?). If published, this will include your full peer review and any attached files.

Reviewer #1: No

Reviewer #2: No

---

## [Editor Report · Acceptance letter]

18 Sep 2023

PONE-D-23-15727R1 

Influence of scar age, laser type and laser treatment intervals on adult burn scars: A systematic review and meta-analysis. 

Dear Dr. Ma:

I'm pleased to inform you that your manuscript has been deemed suitable for publication in PLOS ONE. Congratulations! Your manuscript is now with our production department. 

Kind regards, 

on behalf of

Dr. Ahmed Mustafa Rashid 

Academic Editor

PLOS ONE